# Forward Target Propagation: A Forward-Only Approach to Global Error Credit Assignment via Local Losses

## Abstract

Training neural networks has traditionally relied on backpropagation (BP), a gradient-based algorithm that, despite its widespread success, suffers from key limitations in both biological and hardware perspectives. These include backward error propagation by symmetric weights, non-local credit assignment, update locking, and frozen activity during backward passes. We propose Forward Target Propagation (FTP), a biologically plausible and computationally efficient alternative that replaces the backward pass with a second forward pass. FTP estimates layer-wise targets using only feedforward computations, eliminating the need for symmetric feedback weights or learnable inverse functions, hence enabling modular and local learning. We evaluate FTP on fully connected networks, CNNs, and RNNs, demonstrating accuracies competitive with BP on MNIST, CIFAR-10, and CIFAR-100, as well as effective modeling of long-term dependencies in sequential tasks. FTP shows improved robustness under quantized low-precision and emerging hardware constraints while also demonstrating substantial efficiency gains over other biologically inspired methods such as target propagation variants and forward-only learning algorithms. With its minimal computational overhead, forward-only nature, and hardware compatibility, FTP provides a promising direction for energy-efficient on-device learning and neuromorphic computing.

## 1 Introduction

Backpropagation (BP) has been the foundational algorithm for training neural networks, driving the success of deep learning in tasks such as image recognition, language modeling, and decision-making (Rumelhart et al., 1986; Vaswani et al., 2017; Brown et al., 2020). Despite its proven effectiveness, BP faces key limitations in biological plausibility and hardware compatibility. A major issue is its reliance on symmetric weight transport between forward and backward passes, which conflicts with biological learning, where synaptic updates are not symmetric (Whittington & Bogacz, 2019). Moreover, on edge devices and emerging analog in-memory hardware such as resistive random access memory (RRAM) and phase-change memory (PCM) crossbars, symmetric transport poses further challenges (Yi et al., 2023; 2022; Shafiee et al., 2016). Each training iteration requires weight matrices to be written and verified in transposed form for backward error propagation. However, achieving perfect forward–backward symmetry in analog hardware is often unfeasible (Li et al., 2018), leading to performance degradation. This issue is amplified in edge applications, where low-bit precision constraints worsen the problem (Yang & Sze, 2019). Additionally, BP propagates a global error signal backward through multiple layers, unlike the localized learning signals of biological neural systems (Lillicrap et al., 2020). Its weight updates depend on information from distant layers, implying neurons must access far-off signals, a non-locality absent in biology. BP also suffers from vanishing and exploding gradients, where values become too small for meaningful updates or too large, destabilizing training, especially in deep and recurrent networks (Bengio et al., 1994). Furthermore, BP assumes layer-wise processing and exact error computation, whereas biological systems learn in a noisier, more heuristic manner rather than through precise gradients (Crick, 1989; Scellier et al., 2023). These challenges have driven interest in biologically plausible alternatives that better capture how the brain may perform credit assignment (Lillicrap et al., 2020; 2016; Bengio, 2014; Hinton, 2022; Millidge et al., 2021). This paradigm is worth exploring because the brain

achieves remarkable efficiency and adaptability through local learning rules, providing valuable insights for developing energy-efficient learning algorithms.

In this work, we introduce Forward Target Propagation (FTP), a biologically inspired, forward-only learning framework that addresses key limitations of backpropagation and demonstrate its theoretical basis, empirical competitiveness, efficiency across architectures and tasks, scalability among forward learning approaches, and compatibility with emerging hardware. Our key contributions are summarized as:

- We introduce Forward Target Propagation (FTP), a biologically plausible, forward-only learning algorithm that enables local credit assignment without relying on backward error propagation or symmetric weight transport.

- We evaluate FTP across fully connected, convolutional, and recurrent neural networks on image classification and multivariate time-series forecasting tasks, demonstrating competitive accuracy with backpropagation while being more efficient than other biologically inspired methods.

- We present theoretical justification and empirical evidence showing strong alignment between FTP and backpropagation, measured via the angular similarity between their respective gradient directions during training.

- We assess FTP's robustness in low-precision and noisy hardware settings and benchmark its efficiency in TinyML scenarios, thereby showing that FTP achieves near-BP-level efficiency while outperforming other biologically plausible algorithms. This highlights FTP's suitability for edge and neuromorphic computing.

## 2 RELATED WORK

### 2.1 CONVENTIONAL AND BIOLOGICALLY PLAUSIBLE METHODS

Backpropagation (BP) has long been the backbone of neural network training, relying on forward and backward passes to compute error gradients. In the forward pass, inputs propagate through the network to the output layer, where prediction errors are computed against target values. The backward pass then sends error derivatives through the network using the same weights used in the forward pass, and the outer product of activity vectors from both passes forms the gradient matrices of the global objective. Despite its success, BP is often criticized for its biological implausibility, raising questions about whether such mechanisms exist in real biological systems (Rumelhart et al., 1986). To address the limitations of backpropagation, several biologically inspired learning algorithms have been proposed. Feedback Alignment (FA) (Lillicrap et al., 2016) replaces symmetric feedback with fixed random weights, demonstrating that networks can learn without exact weight mirroring. Direct Feedback Alignment (DFA) extends this idea by projecting output errors directly to hidden units via random matrices (Nøkland, 2016). A simplified variant, Direct Random Target Projection (DRTP), further removes the need for global error signals by using target labels to update hidden layers (Frenkel et al., 2021). Although these methods mitigate issues such as weight transport and update locking, they remain limited in biological plausibility due to their reliance on global error signals and frozen network dynamics. Moreover, DRTP has shown reduced accuracy and poor scalability compared to more recent approaches.

Equilibrium Propagation offers a biologically inspired alternative to backpropagation by computing gradients through equilibrium states in energy-based models (Scellier & Bengio, 2017; Scellier et al., 2023; Yi et al., 2023). Target Propagation (TP) and Difference Target Propagation (DTP) assign credit via layer-wise targets rather than global error gradients, thus avoiding a full backward pass (Bengio, 2014; Lee et al., 2015; Meulemans et al., 2020; Ernoult et al., 2022). While DTP improves target estimation with correction terms, both methods become costly in deep networks. Local Representation Alignment (LRA-E) refines this idea by computing local targets from downstream errors through fixed feedback pathways, removing the need for gradients or symmetric weights (Ororbia & Mali, 2019). Its recursive variant, rec-LRA, scales this mechanism by aligning internal representations layer-by-layer, enabling parallel updates in deep architectures (Ororbia et al., 2023). Though effective on large datasets, both methods introduce additional computational and memory overhead and need careful tuning for stability. Fixed Weight DTP (FWDTP) simplifies DTP with fixed random feedback matrices but suffers performance degradation in deeper networks (Shibuya et al., 2023).

These biologically inspired training schemes aim to match BP's performance while mitigating the weight transport problem. However, challenges such as non-local credit assignment, frozen activity, update locking, and high computational cost remain, underscoring the need for further exploration in this domain.

## 2.2 FORWARD LEARNING METHODS

Forward learning methods offer a biologically plausible alternative to backpropagation by avoiding explicit backward passes and instead utilizing feedforward mechanisms to propagate learning signals. PEPITA is one such forward learning method (Dellaferrera & Kreiman, 2022). It avoids backpropagation by performing a second forward pass where inputs are modulated by output errors through a fixed random matrix (Dellaferrera & Kreiman, 2022). Weight updates come from the difference between standard and modulated activations, enabling local learning without symmetric weights. This removes the weight symmetry constraint of backpropagation but faces the issue of scalability beyond 2 hidden layers. Forward-Forward Learning (FF), introduced by Hinton (Hinton, 2022), eliminates backward error propagation by training solely with forward passes. It compares neuron activations under "positive" and "negative" data, reinforcing those aligned with the desired output. By avoiding backward gradients, FF addresses weight transport and frozen activity. Yet the absence of top-down influence limits inter-layer coordination, partly alleviated by adding recurrent connections at the expense of higher computational complexity. FF also struggles to scale and underperforms beyond simple fully connected (FC) networks, such as convolutional neural networks (CNNs) (Lv et al., 2025). Predictive Forward-Forward Learning (PFF) extends FF with a generative circuit that predicts neural activity to guide local updates (Ororbia & Mali, 2023). Although more biologically plausible and less reliant on error feedback than predictive coding, PFF incurs extra cost from repeated prediction steps.

Despite these advancements, forward learning methods remain limited in efficiency, performance, and scalability. Although they address key biological implausibilities of BP such as symmetric weight transport and global error gradients, they often introduce new issues, including greater computational demands and poor scaling in deeper networks.

## 3 FORWARD TARGET PROPAGATION

### 3.1 OVERVIEW OF ALGORITHM

Forward Target Propagation (FTP) is a novel algorithm that replaces the backward pass in neural networks with a second forward pass to compute learning signals. At the core of FTP lies the concept of layer-wise targets similar to target propagation algorithms (Bengio, 2014; Lee et al., 2015; Meulemans et al., 2020; Shibuya et al., 2023), which define the desired activations that each layer should produce during learning. The process begins with a standard forward pass (first forward pass), where the input data $X$ is propagated through the network to generate the output activation $h_L$ based on the current weights. After obtaining this prediction, the target for the first hidden layer, denoted as $\tau_1$, is estimated using a difference-corrected projection. Specifically, the final-layer output $h_L$ and the ground-truth label $y$ are independently projected through a fixed random matrix $G$, initialized with zero mean and small standard deviation. The resulting estimate is computed as:

$$\tau_1 = \sigma(Gy) - \sigma(Gh_L) + h_1, \tag{1}$$

where $\sigma(\cdot)$ is the element-wise nonlinearity, and $h_1$ is the activation of the first hidden layer from the first forward pass. The estimated targets then serve as inputs for the second forward pass, during which the same feedforward weights used in the first forward computation are reused to propagate the target signal and estimate layer-wise targets. Finally, the weights are updated by minimizing a local loss between activations and targets, thereby enabling layer-wise learning without backward gradients or symmetric feedback.

This forward-only update features a pipelined training by overlapping minibatches: once the first layer's weights are updated and the next layer's target is being computed, the following minibatch can begin its first forward pass. In contrast, BP requires all layers to complete backward gradient computation and weight updates before processing the next minibatch, resulting in strict update locking. FTP thus allows overlapping computation of minibatches, hereby solves update locking

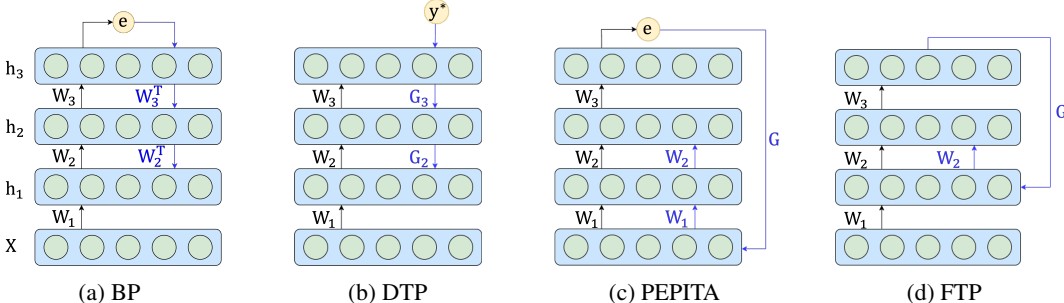

(a) BP        (b) DTP        (c) PEPITA        (d) FTP

Figure 1: Configuration of various learning methods: a) BP, b) DTP, c) PEPITA, and d) FTP. Black arrows represent forward paths, while blue arrows indicate error/target paths. Each $\boldsymbol{W}$ denotes the learnable weight matrices of hidden layers, whereas the top-down feedback path includes **learnable** matrices ($\boldsymbol{G}_3, \boldsymbol{G}_2$ for DTP) or **randomly initialized and frozen** matrix $\boldsymbol{G}$ (for PEPITA and FTP).

partially and can lead to improved wall-clock efficiency over a full training cycle than BP. For comparison, the configuration of various learning methods- BP, DTP, PEPITA, and FTP, is shown in Figure 1.

### 3.2 TARGET ESTIMATION AND LEARNING

In a feedforward neural network, the process begins with input data $\boldsymbol{X}$ and corresponding output labels $\boldsymbol{y}$. The activation values for the input layer (layer 0) are given by: $\boldsymbol{h}_0 = \boldsymbol{X}$. For each subsequent layer $i$ (where $i = 1, 2, \ldots, L$, and $L$ is the final layer), the activation values are computed as:

$$\boldsymbol{h}_i = \sigma(\boldsymbol{W}_i \boldsymbol{h}_{i-1}) \tag{2}$$

where $\boldsymbol{W}_i$ is the weight matrix for layer $i$, $\boldsymbol{h}_{i-1}$ is the activation from the previous layer, and $\sigma$ is the activation function applied element-wise (e.g., *tanh*, sigmoid).

FTP assigns each layer a target via forward propagation of the initial target signal. We estimate the first target $\boldsymbol{\tau}_1$, which is the target for the first hidden layer, using a fixed random matrix $\boldsymbol{G}$, through a difference-corrected projection that contrasts the label and current output in a transformed space:

$$\boldsymbol{\tau}_1 = \sigma(\boldsymbol{G}\boldsymbol{y}) - \sigma(\boldsymbol{G}\boldsymbol{h}_L) + \boldsymbol{h}_1 \tag{3}$$

This formulation provides a top-down feedback mechanism to estimate layer-wise targets without relying on the global error gradient propagation. The target for the first hidden layer from Equation (3) reflects how its activation should shift to ultimately produce the correct output, $h_L \approx y$. Although $h_1 \in \mathbb{R}^{d_1}$ and $h_L, y \in \mathbb{R}^{d_L}$, the fixed random projection $G$ maps $h_L, y$ into $\mathbb{R}^{d_1}$. The difference $\sigma(Gy) - \sigma(Gh_L)$ captures the directional discrepancy between the desired output and current prediction in the projected space. Adding this correction to $h_1$ yields a target $\tau_1$ that nudges the first hidden layer toward a representation more likely to result in $h_L \approx y$ after forward propagation.

To transmit this signal to deeper layers, i.e., $i = 2, 3, \ldots, L-1$, we propagate targets recursively using the same feedforward weights used in the first forward pass:

$$\boldsymbol{\tau}_i = \sigma(\mathbf{W}_i \boldsymbol{\tau}_{i-1}), \tag{4}$$

Each layer's weights are then updated by minimizing a local loss that encourages alignment between activation and target:

$$\mathcal{L}_i = \begin{cases} \|\boldsymbol{h}_i - \boldsymbol{\tau}_i\|_2^2 & \text{if } i < L \\ \mathcal{L} & \text{if } i = L, \end{cases} \tag{5}$$

where $\mathcal{L}$ is the global loss (e.g., cross-entropy, mean-squared loss) used at the final layer.

The FTP weight update is obtained by performing standard gradient descent on the local loss $\mathcal{L}_i$, leading to:

$$\Delta\mathbf{W}_i = -\eta\nabla_{\mathbf{W}_i}\mathcal{L}_i \tag{6}$$

where $\eta$ is the learning rate. This update arises directly from minimizing the layer-wise squared loss between activation, $h_i$, and target $\tau_i$.

This formulation demonstrates that FTP enables weight updates using only locally available information, where each update implicitly reflects a top-down influence from the global target through fixed

projections. FTP also eliminates the need to compute the first hidden layer's activation from the input level in second forward pass, unlike other forward-learning algorithms that do so using modulated inputs or negative image samples (Dellaferrera & Kreiman, 2022; Hinton, 2022).

PEPITA, for example, computes an explicit global error at the output layer and projects it back to the input using a fixed feedback matrix. A second forward pass with perturbed input induces a modified hidden activation, which is compared to the original activations to drive learning. This requires explicit error computation and assumes sensory neurons can access and respond to internal errors- an implausible assumption, as early sensory neurons act as feedforward encoders without receiving task-level feedback (Gilbert & Li, 2013). Moreover, explicit error signaling is biologically unsupported, as noted by Guerguiev et al. (Guerguiev et al., 2017), who argue for dendritic compartmentalization as the mechanism for integrating feedforward and feedback signals. In contrast, FTP generates target activations for the first hidden layer using a feedback mapping from the model's output and label, avoiding explicit error computation and input perturbation. This aligns with cortical pyramidal neuron structure, where basal dendrites receive feedforward inputs and apical dendrites receive feedback, supporting local plasticity through dendritic mismatch (Guerguiev et al., 2017). Furthermore, as shown in Appendix C, this mechanism introduces the global error $e$ through this forward target propagation, which produces updates that align closely with the BP gradients and exhibit similarity to Gauss-Newton directions. The complete training procedure is outlined in Algorithm 1.

---

**Algorithm 1:** FTP

**Given:** Input $X$, label $y$
$h_0 = X, \tau_L = y$
\# First Forward Pass
**for** $i = 1$ to $L$ **do**
  $h_i = \sigma(W_i h_{i-1})$
**end for**
\# Estimate First Target
$\tau_1 = \sigma(G \tau_L) - \sigma(G h_L) + h_1$
\# Second Forward Pass
**for** $i = 1$ to $L$ **do**
  **if** $1 < i < L$ **then**
    $\tau_i = \sigma(W_i \tau_{i-1})$
  **end if**
  $\Delta W_i = \frac{\partial \mathcal{L}_i}{\partial W_i}$
**end for**

---

## 4 RESULTS AND DISCUSSION

### 4.1 METHODS

We evaluated the performance of FTP against conventional BP and other biologically plausible algorithms such as PEPITA and DTP across two task categories: image classification using FC and CNN architectures, and time-series forecasting using recurrent neural network (RNN).

All models were trained using cross-entropy loss for classification tasks and mean squared error for time-series forecasting, optimized by SGD with momentum. For consistency across evaluations, each model family (FC, CNN, and RNN) was implemented using a consistent architecture across all corresponding datasets. Architectural specifications, activation functions, initialization schemes, and training schedules are provided in the Appendix D. Furthermore, we evaluate the impact of initialization choice of feedback matrix, $G$ which is elaborated in Appendix E.

### 4.2 FTP FOR IMAGE CLASSIFICATION

We evaluated FTP on FC and CNN networks for image classification. FC models were tested on MNIST, FMNIST, and CIFAR-10, while CNNs were evaluated on MNIST, CIFAR-10, and CIFAR-100. Results are summarized in Table 1. Across FC architectures, FTP consistently outperforms DTP and achieves accuracy comparable to BP and PEPITA, but with significantly lower computational cost as shown in Table 2. Similarly, in CNNs, FTP performs competitively, demonstrating its ability to learn spatial features effectively.

Table 1: Test accuracy [%] achieved by BP, DTP, PEPITA, and FTP in the experiments for fully connected (FC) networks and convolutional neural networks (CNNs).

| Algorithm | FC Networks | | | CNNs | | |
|---|---|---|---|---|---|---|
| | MNIST | FMNIST | CIFAR-10 | MNIST | CIFAR-10 | CIFAR-100 |
| BP | $98.27_{\pm 0.08}$ | $89.10_{\pm 0.12}$ | $55.31_{\pm 0.26}$ | $98.74_{\pm 0.05}$ | $64.88_{\pm 0.18}$ | $33.83_{\pm 0.26}$ |
| DTP | $96.51_{\pm 0.34}$ | $85.87_{\pm 0.41}$ | $48.67_{\pm 0.19}$ | $97.23_{\pm 0.22}$ | $52.76_{\pm 0.27}$ | $23.51_{\pm 0.58}$ |
| PEPITA | $98.05_{\pm 0.11}$ | $88.41_{\pm 0.14}$ | $52.45_{\pm 0.28}$ | $98.41_{\pm 0.24}$ | $56.17_{\pm 0.62}$ | $26.77_{\pm 0.87}$ |
| FTP | $97.98_{\pm 0.25}$ | $87.24_{\pm 0.21}$ | $52.57_{\pm 0.37}$ | $98.28_{\pm 0.35}$ | $56.32_{\pm 0.82}$ | $26.84_{\pm 1.13}$ |

Table 2: MACs (millions) for BP, DTP, PEPITA, and FTP on FC networks, along with percentage change in MAC (%) with respect to BP.

| Dataset | | BP | DTP | PEPITA | FTP |
|---|---|---|---|---|---|
| MNIST | MAC (mil.) | 2.00 | 2.94 | 2.81 | **2.02** |
| | MAC (%) | 0.00% | 66% | 41% | **1%** |
| CIFAR-10 | MAC (mil.) | 6.69 | 9.97 | 9.86 | **6.71** |
| | MAC (%) | 0% | 35% | 32% | **0%** |
| CIFAR-100 | MAC (mil.) | 6.72 | 10.01 | 10.18 | **6.93** |
| | MAC (%) | 0% | 36% | 34% | **3%** |

Table 2 also shows that FTP requires 30–60% fewer multiply-accumulate (MAC) operations per input sample than other bio-plausible methods such as PEPITA and DTP, while remaining close to BP in computational cost. This highlights FTP's practical efficiency in such shallow-to-moderately deep architectures, making it a compelling forward-only alternative for both dense and convolutional models, especially in resource-constrained settings.

## 4.3 FTP in Capturing Long-Term Dependency

We evaluated FTP on three standard time-series forecasting benchmarks, Electricity, METR-LA (Traffic), and Solar Energy (Lai et al., 2017), using a RNN trained to predict the $25^{th}$ time sample from the previous 24. As in FC and CNN settings, FTP estimates targets using a difference-corrected projection through a fixed random matrix. We evaluated model performance using two standard time-series metrics: Root Relative Squared Error (RRSE) and the Pearson correlation coefficient (CORR). These metrics are defined in Appendix F.

RNNs are inherently deep due to their temporal unrolling and Table 3 shows that FTP consistently performs well across all datasets in RNNs. FTP has higher correlation coefficient than BP on all three benchmarks and remains competitive in RRSE. On METR-LA, FTP achieves both the lowest RRSE and highest CORR. These results suggest that FTP is particularly effective at capturing temporal structure and underlying patterns in time-series data, a crucial aspect of real-world forecasting tasks. This highlights FTP's potential as a biologically plausible alternative to BPTT, with improved robustness and compatibility for analog or on-device learning environments, which is discussed in Section 4.6.

Table 3: Performance of RNN-based models on time-series datasets using BP, PEPITA, and FTP. ↑ (↓) indicates that higher (lower) values are better.

| Method | Electricity | | METR-LA | | Solar | |
|---|---|---|---|---|---|---|
| | RRSE↓ | CORR↑ | RRSE↓ | CORR↑ | RRSE↓ | CORR↑ |
| BP (BPTT) | 0.1059 | 0.9302 | 0.4680 | 0.8750 | 0.1161 | 0.9919 |
| PEPITA-RNN | 0.1214 | 0.9929 | 0.4419 | 0.8967 | 0.1170 | 0.9932 |
| **FTP-RNN** | 0.1219 | 0.9935 | 0.4398 | 0.8987 | 0.1177 | 0.9931 |

## 4.4 Scalability of FTP

To demonstrate scalability of FTP, we evaluated both deep FC networks (all hidden layers of 1024 units) and convolutional architectures. Details on architecture choice are described in Appendix G. As evident from Table 4, across datasets, BP achieves the highest accuracy, but FTP consistently remains close to BP, while being far more stable and better than PEPITA, which collapses with depth. Prior work has shown that forward-only learning schemes without extra learnable parameters, such as FF, were restricted to at most four hidden layers (Hinton, 2022), while PEPITA was originally limited to one hidden layer (Dellaferrera & Kreiman, 2022) and required auxiliary techniques such as normalization or weight decay to stabilize deeper FC models (Srinivasan et al., 2023). Potential scalability for CNNs using FTP is discussed in Appendix G. Our results demonstrate that FTP can train both deeper FC and convolutional networks without additional learning (Gong et al., 2025) or stabilization techniques, and produces predictions directly in a single forward pass during inference, unlike FF, which requires testing all labels explicitly for each sample.

Table 4: Test accuracy (Mean ± Std) of BP, PEPITA, and FTP across datasets and hidden layer depths for FC networks. Here, L = Total number of layers in network.

| Dataset | Algorithm | L = 3 | L = 4 | L = 5 | L = 6 |
|---------|-----------|-------|-------|-------|-------|
| MNIST | BP | 98.48±0.09 | 98.55±0.05 | 98.60±0.06 | 98.60±0.05 |
| | PEPITA | 98.19±0.03 | 94.63±0.21 | 89.12±0.35 | 81.71±0.80 |
| | **FTP** | **98.36**±0.07 | **98.15**±0.11 | **98.24**±0.08 | **98.20**±0.07 |
| CIFAR-10 | BP | 57.83±0.08 | 58.06±0.19 | 57.86±0.39 | 57.65±0.37 |
| | PEPITA | 52.46±0.17 | 52.78±0.56 | 44.88±0.19 | 32.72±0.36 |
| | **FTP** | **53.88**±0.24 | **54.81**±0.18 | **54.72**±0.16 | **54.93**±0.13 |
| CIFAR-100 | BP | 30.08±0.17 | 29.31±0.28 | 28.50±0.11 | 28.55±0.29 |
| | PEPITA | 24.80±0.32 | 18.05±0.16 | 1.20±0.11 | 6.57±0.48 |
| | **FTP** | **26.27**±0.27 | **27.12**±0.20 | **27.85**±0.36 | **28.35**±0.54 |

## 4.5 ALIGNMENT OF FTP WITH BACKPROPAGATION

Gradient alignment measures how closely alternative learning rules match BP's parameter update directions. This metric is particularly useful in the context of biologically plausible algorithms (Lillicrap et al., 2016; Nøkland, 2016; Shervani-Tabar & Rosenbaum, 2023), where exact gradient computation is often replaced by approximate or learned feedback pathways. We quantify the alignment by computing the cosine angle between flattened gradient vectors of an approximate method and BP throughout training. A smaller angle indicates a closer match in update directions, which often correlates with improved learning performance and better convergence. To analyze the gradient alignment of FTP with BP, we use a fully connected network with two hidden layers, trained on MNIST for 100 epochs, and average the results over 10 random seeds. A scaling parameter $\gamma$, introduced in detail below, is set to 1 at this point, which corresponds to the standard FTP target. The alignment angle for the output layer always remains at zero during training, since FTP and BP share identical gradient expressions for the final layer. As shown in Figure 2a, the alignment angle for both hidden layers starts near $90°$, indicating no initial alignment between the FTP and BP gradients. This is expected, as the fixed random feedback matrix $\boldsymbol{G}$ used in FTP is uncorrelated with the forward weights at initialization, unlike BP, which computes exact gradients via the chain rule. As training progresses, however, the angle steadily decreases, which reflects that FTP's updates increasingly align with those of BP.

To understand how the magnitude of the error signal affects alignment, we introduce a hyperparameter $\gamma$ that scales the FTP target as

$$\tau_1 = \gamma(\sigma(\boldsymbol{G}\boldsymbol{y}) - \sigma(\boldsymbol{G}\boldsymbol{h}_L)) + \boldsymbol{h}_1 \tag{7}$$

Setting $\gamma = 1$ yields the standard FTP formulation. We evaluated its effect over the range [0.1, 1.5], and here we report results for $\gamma = 0.5$ and $\gamma = 1.5$ in Figure 2c and 2d to highlight the impact of error signal scaling. Each setting was evaluated for 10 random seeds, and only the mean alignment is shown for visual clarity. When $\gamma = 0.5$, we observe stronger alignment in both hidden layers, compared to the case of $\gamma = 1$. Conversely, increasing $\gamma$ to 1.5 degrades the alignment. These results indicate that $\gamma$ acts as a regularizer, which governs error-signal strength and directional alignment with BP. When $\gamma$ is large, the FTP target shift becomes overly aggressive, potentially pushing the hidden representations too far from their initial state, which can destabilize learning and lead to misaligned updates. In contrast, a smaller $\gamma$ produces more conservative target shifts, allowing the network to adjust gradually in directions that are more naturally aligned with the forward pathway and ultimately closer to the BP gradients.

In addition to gradient alignment, we analyze the structural alignment between the product of forward weights (i.e., $\boldsymbol{W}_2^T \boldsymbol{W}_3^T$) and the projection matrix $\boldsymbol{G}$, following FA(Lillicrap et al., 2016). We flatten them and compute the angle between the flattened vectors throughout training, which is depicted in Figure 2b. Initially, the angle is close to $90°$, reflecting a random orientation. Over time, the angle steadily decreases, indicating that FTP promotes consistent forward–backward correlations. This behavior contrasts with the anti-alignment in PEPITA (Dellaferrera & Kreiman, 2022), and supports the hypothesis (Lillicrap et al., 2016) that biologically plausible learning rules such as FTP naturally give rise to structured and coordinated representations.

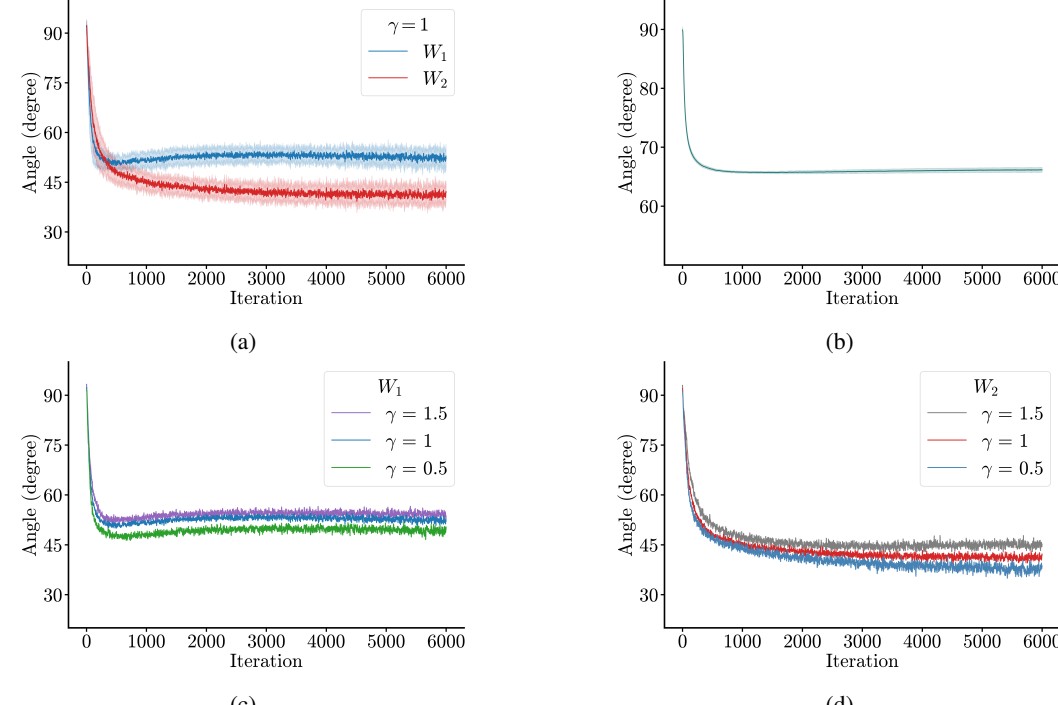

Figure 2: Evolution of alignment during training: (a) Alignment between gradient directions from FTP and BP with $\gamma = 1$; (b) Alignment between products of forward weight matrices ($\boldsymbol{W}_2^T \boldsymbol{W}_3^T$) and the feedback matrix $\boldsymbol{G}$; (c,d) Alignment of gradients for $\boldsymbol{W}_1$ and $\boldsymbol{W}_2$ under varying $\gamma$.

### 4.6 PERFORMANCE OF FTP IN EMERGING HARDWARE AND EDGE DEVICES

BP relies on symmetry between forward and backward weights, which is difficult to maintain on emerging analog accelerators such as RRAM and other non-volatile memories, due to non-idealities including noise, programming errors, and variability. These asymmetries disrupt gradient flow and significantly degrade BP performance under low-bit precision. We demonstrate in Appendix J how the performance of BP is susceptible to such non-ideal conditions.

Programming errors, a key source of non-ideality, accumulate differently in FTP and BP. In FTP, the feedback matrix $\boldsymbol{G}$ is randomly initialized and programmed only once at the start of training. Any programming error introduced then remains embedded in $\boldsymbol{G}$ throughout the training process. Since FTP reuses the same forward matrices for both activations and layer-wise targets, any programming errror in matrices affects both pathways in consistent manner, making it inherently robust to hardware-induced asymmetry and stable even in noisy, low-precision settings. In contrast, BP requires continual reprogramming of forward and backward matrices–unless complicated bidirectional peripheral circuitry is implemented (Wan et al., 2022). Independent reprogramming of forward and backward matrices often violates the symmetry, especially in analog crossbars (e.g., RRAM, PCM), where these two are stored in physically different memory cells (Yi et al., 2023). Here, *symmetry* refers to the proportion of elements in the backward weight matrix that remain uncorrupted relative to the forward matrix. Reduced symmetry means backward weights increasingly deviate from forward-transpose, introducing new, uncorrelated errors at every step.

To evaluate this effect, we injected additive Gaussian noise into weights to simulate programming errors. The noise level, shown on the x-axis of Figure 3, is parameterized by $\alpha$, scaling the standard deviation of noise distribution as std $= \alpha \times |w|$, where $w$ is the weight being corrupted (Xiao et al., 2023). Thus, larger weights incur proportionally larger noise, with higher $\alpha$ producing stronger perturbations. We also compare against FA, which uses fixed random feedback matrices and avoids reprogramming backward paths in hardware, making it a relevant robustness baseline (see Appendix H for detailed discussion). As shown in Figure 3 and Appendix H, FTP consistently shows improved robustness under programming errors, especially as noise levels increase. Notably, in 4-bit systems, FTP maintains higher accuracy than BP, underscoring its robustness to programming errors in low-precision and non-ideal hardware.

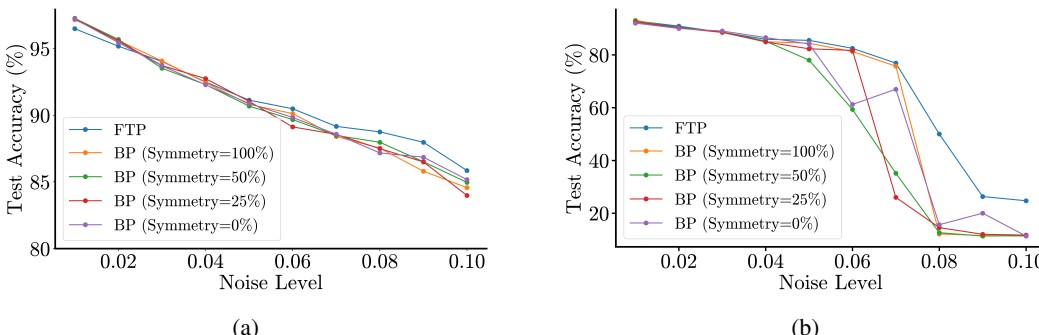

(a)                                    (b)

Figure 3: Performance of FTP and BP when programming errors are considered for (a) 8-bit and (b) 4-bit precision devices.

### 4.7 HARDWARE-AWARE ANALYSIS OF FTP'S EFFICIENCY

To assess FTP's suitability for resource-constrained environments such as TinyML applications, we estimated the number of MAC operations based on dataset sizes and epoch counts aligned with the MLCommons/Tiny Benchmark (MLCommons, 2023), following a methodology similar to Pau & Aymone (2023). This estimation offers a hardware-aware perspective on training efficiency under realistic deployment constraints. As shown in Table 5, FTP consistently achieves lower or comparable MAC counts relative to BP and significantly lower computational overhead compared to other forward-only methods such as FF, PEPITA, and DTP. These reductions in MAC operations imply both lower power consumption and faster training–critical factors for edge devices that operate under tight energy and latency budgets. If pipelined scheduling is enabled, FTP can even be more efficient than BP in overall runtime, as elaborated in Appendix I. Combined with FTP's robustness to noisy gradients and tolerance to analog hardware imperfections, FTP emerges as a promising candidate for on-device learning in resource-constrained systems.

Table 5: Comparison of MAC (millions) for TinyML datasets with various algorithms and percentage change in MAC (%) with respect to BP.

| Learning Method | DS-CNN/SC | | MobileNet/VWW | | ResNet/CIFAR10 | | AE/ToyADMOS | |
|---|---|---|---|---|---|---|---|---|
| | MAC | MAC (%) | MAC | MAC (%) | MAC | MAC (%) | MAC | MAC (%) |
| BP | 7.7 | 0.00% | 22.4 | 0.00% | 37.1 | 0.00% | 0.7 | 0.00% |
| FF | 10.9 | 42.58% | 31.5 | 40.70% | 50.4 | 35.89% | 1.1 | 49.51% |
| PEPITA | 8.0 | 4.25% | 22.9 | 2.47% | 37.6 | 1.35% | 1.2 | 69.10% |
| DTP | 17.1 | 122.7% | 50.7 | 126.7% | 76.5 | 106.1% | 1.5 | 104% |
| **FTP** | **7.9** | **2.51%** | **22.4** | **0.33%** | **37.5** | **0.96%** | **0.9** | **23.03%** |

## 5 CONCLUSION

We have introduced Forward Target Propagation (FTP), a biologically plausible and computationally efficient learning algorithm that serves as a forward-only alternative to backpropagation. FTP achieves performance comparable to that of BP across a variety of architectures, including visual pattern recognition and long-term temporal modeling. A key feature of FTP is its ability to assign global credit through purely local losses, thereby eliminating the need for weight symmetry, non-local objective function, and backward signal propagation. Among recent bio-inspired algorithms and forward-only learning algorithms, our results demonstrate that FTP shows comparable performance and better scalability along with greater computational efficiency–yielding lower multiply-accumulate (MAC) operations than FF, PEPITA, and DTP. FTP also exhibits resilience to low-bit precision and noisy hardware conditions, where conventional BP tends to degrade. These advantages position FTP as an effective solution for deployment in TinyML and neuromorphic systems. Current study focuses on small to medium-scale architectures, and extending FTP to deeper networks and larger datasets remains an important direction for future work. Furthermore, real hardware validation and exploration of FTP's ability to support more complex learning mechanisms such as attention in large language models would offer valuable insights into its scalability and broader applicability. While further development is needed to fully realize its potential at scale, our findings establish FTP as a robust, energy-efficient, and biologically plausible learning framework for embedded AI systems and neuromorphic computing.

REPRODUCIBILITY STATEMENT

The FTP algorithm and update rules are fully described in Section 3, with theoretical analysis and proofs provided in Appendix C. Experimental setups, including datasets, architectures, hyperparameters, and training schedules, are detailed in Appendices D, E and F. Codes used in this work have been shared as supplementary materials in anonymized form to allow independent verification of our results.

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

## A   BROADER IMPACT OF OUR WORK

As AI systems, particularly large language models (LLMs), grow in scale and deployment breadth, the energy consumption has become increasingly critical. Our work proposes Forward Target Propagation (FTP), a biologically inspired, forward-only learning algorithm that offers significant advantages in efficiency, robustness, and hardware compatibility compared to standard backpropagation (BP). These characteristics make FTP particularly relevant for enabling sustainable training and deployment of large-scale models in both cloud and edge settings. Below, we outline key broader impacts of our approach:

**Compatibility with Quantization and Noisy Hardware:** Emerging memory technologies such as Resistive Random Access Memory (RRAM) and Phase-Change Memory (PCM) offer compact, analog, and energy-efficient platforms for neural computation, but suffer from non-idealities such as stochastic noise, variability and limited precision. Unlike BP, which requires high-precision symmetric weight updates, FTP is inherently robust to such hardware imperfections due to its forward-only and fixed-feedback structure. Our experimental results demonstrate that FTP consistently outperforms BP under low-bit precision and noisy conditions, making it particularly well suited for deployment on in-memory computing systems.

**Enabling Sustainable, On-Device Intelligence:** By aligning algorithmic design with the constraints and strengths of emerging and edge hardware, FTP opens the door to efficient on-device learning in applications such as autonomous sensors, wearable devices, and edge robotics. Its robustness and low-power profile help advance the goal of sustainable and accessible AI, particularly in scenarios where cloud computation is infeasible due to latency, privacy, or energy constraints.

## B   BIOLOGICAL PROPERTIES OF FTP

Table 6 compares various learning rules across key criteria relevant to biological plausibility. A checkmark (✓) indicates that the corresponding issue is addressed (i.e., the property is satisfied), while a cross (✗) indicates that the issue remains unsolved. The tilde symbol (∼) denotes a partial solution. Notably, FTP satisfies all the listed criteria, with a partial resolution of update locking.

Table 6: Comparison of learning rules based on key metrics.

| Learning Rule | Forward Only | Weight Symmetry | Local | Activity Freezing | Update Unlocked |
|---|---|---|---|---|---|
| BP | ✗ | ✓ | ✗ | ✗ | ✗ |
| DTP | ✗ | ✓ | ✓ | ✗ | ✗ |
| PEPITA | ✓ | ✓ | ✓ | ✓ | ∼ |
| FF | ✓ | ✓ | ✓ | ✓ | ✓ |
| FTP | ✓ | ✓ | ✓ | ✓ | ∼ |

## C  ANALYSIS OF FTP IN NETWORK WITH TWO HIDDEN LAYERS

We consider a neural network with two hidden layers with activation $\sigma(\cdot)$. In Section C.1, we proved that (under some conditions) the weight update direction for the hidden layers in FTP is within $90°$ of that of BP. In Section C.2, we showed that the update direction for the hidden neurons in FTP is aligned with the Gauss-Newton direction. We followed the structure of the proofs from (Lillicrap et al., 2016).

For an input $\boldsymbol{x} \in \mathbb{R}^{d_x}$, we have

$$\boldsymbol{h}_1 = \sigma(\boldsymbol{W}_1 \boldsymbol{x}) \tag{8}$$

$$\boldsymbol{h}_2 = \sigma(\boldsymbol{W}_2 \boldsymbol{h}_1) \tag{9}$$

$$\boldsymbol{h}_3 = \sigma(\boldsymbol{W}_3 \boldsymbol{h}_2) \tag{10}$$

where $\boldsymbol{h}_3 \in \mathbb{R}^{d_y}$ is the output, and $\boldsymbol{h}_1 \in \mathbb{R}^{d_1}, \boldsymbol{h}_2 \in \mathbb{R}^{d_2}$ are hidden neurons. In forward target propagation (FTP), the losses to train the weight matrices $\boldsymbol{W}_1, \boldsymbol{W}_2$ and $\boldsymbol{W}_3$ are as follows:

$$\mathcal{L}_1 = \frac{1}{2} \|\operatorname{stop\_grad}(\boldsymbol{\tau}_1) - \boldsymbol{h}_1\|^2 \tag{11}$$

$$\mathcal{L}_2 = \frac{1}{2} \|\operatorname{stop\_grad}(\boldsymbol{\tau}_2) - \boldsymbol{h}_2\|^2 \tag{12}$$

$$\mathcal{L}_3 = \frac{1}{2} \|\boldsymbol{y} - \boldsymbol{h}_3\|^2 \tag{13}$$

where $\boldsymbol{y} \in \mathbb{R}^{d_y}$ is the target value at the output layer and $\boldsymbol{\tau}_1, \boldsymbol{\tau}_2$ are target values (suggested by FTP) at the hidden layers according to the following.

$$\boldsymbol{\tau}_1 = \boldsymbol{h}_1 + \sigma(\boldsymbol{G}\boldsymbol{y}) - \sigma(\boldsymbol{G}\boldsymbol{h}_3) \tag{14}$$

$$\boldsymbol{\tau}_2 = \sigma(\boldsymbol{W}_2 \boldsymbol{\tau}_1) \tag{15}$$

$\boldsymbol{G}$ is a $d_1 \times d_y$ projection matrix where $G_{i,j}$ can be distributed according to $\mathcal{N}(0,1)$.

### C.1  ALIGNMENT IN WEIGHT UPDATE DIRECTION BETWEEN FTP AND BP

In this section, we consider the same neural network described above, except for linear activation functions. Denoting error signal as $\boldsymbol{e} = \boldsymbol{y} - \boldsymbol{h}_3$, we have the following gradient direction for each weight matrix.

$$\frac{\partial \mathcal{L}_3}{\partial \boldsymbol{W}_3} = \boldsymbol{e}\boldsymbol{h}_2^T \tag{16}$$

$$\frac{\partial \mathcal{L}_2}{\partial \boldsymbol{W}_2} = (\boldsymbol{\tau}_2 - \boldsymbol{h}_2)\boldsymbol{h}_1^T \tag{17}$$

$$\frac{\partial \mathcal{L}_1}{\partial \boldsymbol{W}_1} = (\boldsymbol{\tau}_1 - \boldsymbol{h}_1)\boldsymbol{x}^T \tag{18}$$

Note the FTP has the same gradient for $\boldsymbol{W}_3$ as in BP. From Equations (14) and (15), we have

$$\boldsymbol{\tau}_1 - \boldsymbol{h}_1 = \boldsymbol{G}\boldsymbol{y} - \boldsymbol{G}\boldsymbol{h}_3 = \boldsymbol{G}\boldsymbol{e}$$

$$\boldsymbol{\tau}_2 - \boldsymbol{h}_2 = \boldsymbol{W}_2\boldsymbol{\tau}_1 - \boldsymbol{W}_2\boldsymbol{h}_1 = \boldsymbol{W}_2(\boldsymbol{\tau}_1 - \boldsymbol{h}_1) = \boldsymbol{W}_2\boldsymbol{G}\boldsymbol{e}$$

Finally, we get the gradient direction for $\boldsymbol{W}_1, \boldsymbol{W}_2$ under FTP as:

$$\frac{\partial \mathcal{L}_2}{\partial \boldsymbol{W}_2} = \boldsymbol{W}_2\boldsymbol{G}\boldsymbol{e}\boldsymbol{h}_1^T \tag{19}$$

$$\frac{\partial \mathcal{L}_1}{\partial \boldsymbol{W}_1} = \boldsymbol{G}\boldsymbol{e}\boldsymbol{x}^T \tag{20}$$

For BP,

$$\frac{\partial \mathcal{L}_2}{\partial \boldsymbol{W}_2} = \boldsymbol{W}_3^T \boldsymbol{e}\boldsymbol{h}_1^T \tag{21}$$

$$\frac{\partial \mathcal{L}_1}{\partial \boldsymbol{W}_1} = \boldsymbol{W}_2^T \boldsymbol{W}_3^T \boldsymbol{e}\boldsymbol{x}^T \tag{22}$$

We hypothesize the following alignment between FTP and BP's gradient direction for any non-zero $\boldsymbol{e}$:

$$\langle \boldsymbol{Ge}, \boldsymbol{W}_2^T \boldsymbol{W}_3^T \boldsymbol{e} \rangle > 0 \tag{23}$$

$$\langle \boldsymbol{W}_2 \boldsymbol{Ge}, \boldsymbol{W}_3^T \boldsymbol{e} \rangle > 0 \tag{24}$$

**Lemma 1.** *If we initialize $\boldsymbol{W}_1$ and $\boldsymbol{W}_3$ with zero entries and $\boldsymbol{W}_2$ being initialized with $\boldsymbol{A}$, then there exist some scalars $s_1, s_{\boldsymbol{W}_1}, s_{\boldsymbol{W}_2}, s_{\boldsymbol{W}_3}$ at every time step (during training using FTP) such that,*

$$\boldsymbol{h}_1 = s_1 \boldsymbol{Gy} \tag{25}$$

$$\boldsymbol{W}_1 = s_{\boldsymbol{W}_1} \boldsymbol{Gy}\boldsymbol{x}^T \tag{26}$$

$$\boldsymbol{W}_2 = \boldsymbol{A}(\boldsymbol{I} + s_{\boldsymbol{W}_2} \boldsymbol{Gy}(\boldsymbol{Gy})^T) \tag{27}$$

$$\boldsymbol{W}_3 = s_{\boldsymbol{W}_3} \boldsymbol{y}(\boldsymbol{AGy})^T \tag{28}$$

*Proof.* Here, we provide the proof by induction. At initialization ($t = 0$) Equations (25) to (28) satisfies with $s_1 = s_{\boldsymbol{W}_1} = s_{\boldsymbol{W}_2} = s_{\boldsymbol{W}_3} = 0$. Now, if this is true for any $t > 0$, then we need to prove this holds for $t + 1$.

$$\boldsymbol{h}_3^{(t)} = \boldsymbol{W}_3^{(t)} \boldsymbol{h}_2^{(t)} \tag{29}$$

$$= s_1^{(t)} s_{\boldsymbol{W}_3}^{(t)} \boldsymbol{y}(\boldsymbol{AGy})^T \boldsymbol{W}_2^{(t)} \boldsymbol{Gy} \tag{30}$$

$$= s_1^{(t)} s_{\boldsymbol{W}_3}^{(t)} \boldsymbol{y}(\boldsymbol{AGy})^T \boldsymbol{A}(\boldsymbol{I} + s_{\boldsymbol{W}_2}^{(t)} \boldsymbol{Gy}(\boldsymbol{Gy})^T)\boldsymbol{Gy} \tag{31}$$

$$(\boldsymbol{AGy})^T \boldsymbol{A}(\boldsymbol{I} + s_{\boldsymbol{W}_2}^{(t)} \boldsymbol{Gy}(\boldsymbol{Gy})^T)\boldsymbol{Gy} = (\boldsymbol{Gy})^T \boldsymbol{A}^T \boldsymbol{A}(\boldsymbol{I} + s_{\boldsymbol{W}_2}^{(t)} \boldsymbol{Gy}(\boldsymbol{Gy})^T)\boldsymbol{Gy}$$

$$= ||\boldsymbol{AGy}||^2 + s_{\boldsymbol{W}_2}^{(t)} ||\boldsymbol{AGy}||^2 ||\boldsymbol{Gy}||^2$$

Now,

$$\boldsymbol{h}_3^{(t)} = s_1^{(t)} s_{\boldsymbol{W}_3}^{(t)} \left( ||\boldsymbol{AGy}||^2 + s_{\boldsymbol{W}_2}^{(t)} ||\boldsymbol{AGy}||^2 ||\boldsymbol{Gy}||^2 \right) \boldsymbol{y} \tag{32}$$

$$= s_3^{(t)} \boldsymbol{y} \quad \text{(denoting } s_3^{(t)} = s_1^{(t)} s_{\boldsymbol{W}_3}^{(t)} \left( ||\boldsymbol{AGy}||^2 + s_{\boldsymbol{W}_2}^{(t)} ||\boldsymbol{AGy}||^2 ||\boldsymbol{Gy}||^2 \right)) \tag{33}$$

$$\boldsymbol{e}^{(t)} = \boldsymbol{y} - \boldsymbol{h}_3^{(t)} = (1 - s_3^{(t)})\boldsymbol{y} \tag{34}$$

$$\boldsymbol{W}_1^{(t+1)} = \boldsymbol{W}_1^{(t)} + \eta_1 \boldsymbol{Ge}^{(t)} \boldsymbol{x}^T$$

$$= s_{\boldsymbol{W}_1}^{(t)} \boldsymbol{Gy}\boldsymbol{x}^T + \eta_1 \boldsymbol{G}(1 - s_3^{(t)})\boldsymbol{y}\boldsymbol{x}^T$$

$$= s_{\boldsymbol{W}_1}^{(t+1)} \boldsymbol{Gy}\boldsymbol{x}^T \quad \left( \text{where } s_{\boldsymbol{W}_1}^{(t+1)} = s_{\boldsymbol{W}_1}^{(t)} + \eta_1(1 - s_3^{(t)}) \right)$$

$$\boldsymbol{W}_2^{(t+1)} = \boldsymbol{W}_2^{(t)} + \eta_2 \boldsymbol{W}_2^{(t)} \boldsymbol{Ge}^{(t)} \boldsymbol{h}_1^{T(t)}$$

$$= \boldsymbol{A} + s_{\boldsymbol{W}_2}^{(t)} \boldsymbol{AGy}(\boldsymbol{Gy})^T + \eta_2 \left( \boldsymbol{A} + s_{\boldsymbol{W}_2}^{(t)} \boldsymbol{AGy}(\boldsymbol{Gy})^T \right) \boldsymbol{G}(1 - s_3^{(t)})\boldsymbol{y}(s_1^{(t)} \boldsymbol{Gy})^T$$

$$= \boldsymbol{A} + (s_{\boldsymbol{W}_2}^{(t)} + \eta_2 s_1^{(t)}(1 - s_3^{(t)}))\boldsymbol{AGy}(\boldsymbol{Gy})^T + \eta_2 s_1^{(t)} s_{\boldsymbol{W}_2}^{(t)}(1 - s_3^{(t)})\boldsymbol{AGy}(\boldsymbol{Gy})^T \boldsymbol{Gy}(\boldsymbol{Gy})^T$$

$$= \boldsymbol{A} + (s_{\boldsymbol{W}_2}^{(t)} + \eta_2 s_1^{(t)}(1 - s_3^{(t)}))\boldsymbol{AGy}(\boldsymbol{Gy})^T + \eta_2 s_1^{(t)} s_{\boldsymbol{W}_2}^{(t)}(1 - s_3^{(t)})\boldsymbol{AGy}||\boldsymbol{Gy}||^2(\boldsymbol{Gy})^T$$

$$= \boldsymbol{A} + s_{\boldsymbol{W}_2}^{(t+1)} \boldsymbol{AGy}(\boldsymbol{Gy})^T$$

Here, $s_{\boldsymbol{W}_2}^{(t+1)} = s_{\boldsymbol{W}_2}^{(t)} + \eta_2 s_1^{(t)}(1 - s_3^{(t)}) + \eta_2 s_1^{(t)} s_{\boldsymbol{W}_2}^{(t)}(1 - s_3^{(t)})\|\boldsymbol{Gy}\|^2$

$$
\begin{aligned}
\boldsymbol{W}_3^{(t+1)} &= \boldsymbol{W}_3^{(t)} + \eta_3 \boldsymbol{e}^{(t)} \boldsymbol{h}_2^{T^{(t)}} \\
&= s_{\boldsymbol{W}_3}^{(t)} \boldsymbol{y}(\boldsymbol{AGy})^T + \eta_3 s_1^{(t)}(1 - s_3^{(t)})\boldsymbol{y}(\boldsymbol{W}_2^{(t)} \boldsymbol{Gy})^T \\
&= s_{\boldsymbol{W}_3}^{(t)} \boldsymbol{y}(\boldsymbol{AGy})^T + \eta_3 s_1^{(t)}(1 - s_3^{(t)})\boldsymbol{y}(\boldsymbol{A}(\boldsymbol{I} + s_{\boldsymbol{W}_2}^{(t)} \boldsymbol{Gy}(\boldsymbol{Gy})^T)\boldsymbol{Gy})^T \\
&= s_{\boldsymbol{W}_3}^{(t)} \boldsymbol{y}(\boldsymbol{AGy})^T + \eta_3 s_1^{(t)}(1 - s_3^{(t)})\left(1 + s_{\boldsymbol{W}_2}^{(t)}\|\boldsymbol{Gy}\|^2\right)\boldsymbol{y}(\boldsymbol{AGy})^T \\
&= s_{\boldsymbol{W}_3}^{(t+1)} \boldsymbol{y}(\boldsymbol{AGy})^T \quad (\text{where } s_{\boldsymbol{W}_3}^{(t+1)} = s_{\boldsymbol{W}_3}^{(t)} + \eta_3 s_1^{(t)}(1 - s_3^{(t)})\left(1 + s_{\boldsymbol{W}_2}^{(t)}\|\boldsymbol{Gy}\|^2\right))
\end{aligned}
$$

$$
\begin{aligned}
\boldsymbol{h}_1^{(t+1)} &= \boldsymbol{W}_1^{(t+1)} \boldsymbol{x} \\
&= \left(\boldsymbol{W}_1^{(t)} + \eta_1 \boldsymbol{Ge}^{(t)} \boldsymbol{x}^T\right)\boldsymbol{x} \\
&= \boldsymbol{h}_1^{(t)} + \eta_1(1 - s_3^{(t)})\boldsymbol{Gy}\|\boldsymbol{x}\|^2 \\
&= s_1^{(t)} \boldsymbol{Gy} + \eta_1(1 - s_3^{(t)})\|\boldsymbol{x}\|^2 \boldsymbol{Gy} \\
&= s_1^{(t+1)} \boldsymbol{Gy} \quad (\text{where } s_1^{(t+1)} = s_1^{(t)} + \eta_1(1 - s_3^{(t)})\|\boldsymbol{x}\|^2)
\end{aligned}
$$

$\square$

**Theorem 1.** *Under the same conditions in Lemma 1, the FTP and BP's gradient direction for $\boldsymbol{W}_1, \boldsymbol{W}_2$ are within $90°$ of each other, i.e.*

$$
\langle \boldsymbol{Ge}, \boldsymbol{W}_2^T \boldsymbol{W}_3^T \boldsymbol{e} \rangle > 0 \tag{35}
$$

$$
\langle \boldsymbol{W}_2 \boldsymbol{Ge}, \boldsymbol{W}_3^T \boldsymbol{e} \rangle > 0 \tag{36}
$$

*Proof.* Note $\langle \boldsymbol{Ge}, \boldsymbol{W}_2^T \boldsymbol{W}_3^T \boldsymbol{e} \rangle = \langle \boldsymbol{W}_2 \boldsymbol{Ge}, \boldsymbol{W}_3^T \boldsymbol{e} \rangle$

From Equations (27) and (28) of Lemma 1, we have

$$
\begin{aligned}
\boldsymbol{W}_3 \boldsymbol{W}_2 &= s_{\boldsymbol{W}_3} \boldsymbol{y}(\boldsymbol{AGy})^T \boldsymbol{A}(\boldsymbol{I} + s_{\boldsymbol{W}_2} \boldsymbol{Gy}(\boldsymbol{Gy})^T) \\
&= s_{\boldsymbol{W}_3} \boldsymbol{y}(\boldsymbol{Gy})^T \boldsymbol{A}^T \boldsymbol{A}(\boldsymbol{I} + s_{\boldsymbol{W}_2} \boldsymbol{Gy}(\boldsymbol{Gy})^T) \\
&= s_{\boldsymbol{W}_3} \boldsymbol{y}(\boldsymbol{Gy})^T \boldsymbol{A}^T \boldsymbol{A} + s_{\boldsymbol{W}_3} s_{\boldsymbol{W}_2}\|\boldsymbol{AGy}\|^2 \boldsymbol{y}(\boldsymbol{Gy})^T \\
&= s_{\boldsymbol{W}_3} \boldsymbol{y}(\boldsymbol{Gy})^T \boldsymbol{A}^T \boldsymbol{A} + s_{3,2} \boldsymbol{y}(\boldsymbol{Gy})^T \quad (\text{denoting } s_{3,2} = s_{\boldsymbol{W}_3} s_{\boldsymbol{W}_2}\|\boldsymbol{AGy}\|^2)
\end{aligned} \tag{37}
$$

Now for the weight matrix of the first hidden layer, $\boldsymbol{W}_1$

$$
\begin{aligned}
\langle \boldsymbol{Ge}, \boldsymbol{W}_2^T \boldsymbol{W}_3^T \boldsymbol{e} \rangle &= (\boldsymbol{Ge})^T (\boldsymbol{W}_3 \boldsymbol{W}_2)^T \boldsymbol{e} \\
&= s_{\boldsymbol{W}_3}(1 - s_3)^2 (\boldsymbol{Gy})^T (\boldsymbol{y}(\boldsymbol{Gy})^T \boldsymbol{A}^T \boldsymbol{A})^T \boldsymbol{y} + s_{3,2}(1 - s_3)^2 (\boldsymbol{Gy})^T (\boldsymbol{y}(\boldsymbol{Gy})^T)^T \boldsymbol{y} \\
&= s_{\boldsymbol{W}_3}(1 - s_3)^2 \|\boldsymbol{AGy}\|^2 \|\boldsymbol{y}\|^2 + s_{3,2}(1 - s_3)^2 \|\boldsymbol{Gy}\|^2 \|\boldsymbol{y}\|^2 > 0 \quad \forall \boldsymbol{y} \neq \boldsymbol{0}
\end{aligned}
$$

The last inequality follows from $s_{\boldsymbol{W}_2}, s_{\boldsymbol{W}_3}$ being positive scalars. (Lemma 1). $\square$

## C.2 RELATION WITH GAUSS-NEWTON UPDATE FOR HIDDEN NEURONS

In linear neural network with two hidden layers, the feedback signal (under FTP) for update in $h_1$ is $Ge$, and we postulate that it is aligned with the Gauss-Newton update signal, $(W_3 W_2)^\dagger$. If we can show there exists a relation such as $sGe = (W_3 W_2)^\dagger e$ with a positive scaler $s$, then we can conclude update direction for the hidden neurons $h_1$ is aligned with the Gauss-Newton direction.

**Theorem 2.** *Under the same conditions in Lemma 1 and $W_2$ being initialized with $A$ such that $A^T A = I$ (requires $d_2 \geq d_1$), there exists a positive scaler $s$ such that*

$$sGe = (W_3 W_2)^\dagger e \tag{38}$$

*Proof.* From Equation (37) with $A^T A = I$,

$$W_3 W_2 = s_{W_3} y(Gy)^T + s_{W_3} s_{W_2} ||Gy||^2 y(Gy)^T$$
$$= s'_{3,2} y(Gy)^T \quad (\text{where } s'_{3,2} = s_{W_3} + s_{W_3} s_{W_2} ||Gy||^2)$$

Showing $sGe = (W_3 W_2)^\dagger e$ is equivalent to $sGy = (y(Gy)^T)^\dagger y$ since we have $e = (1 - s_3)y$ from Equation (34).

$$(y(Gy)^T)^\dagger y = (y^T G^T)^\dagger y^\dagger y$$
$$= (G^T)^\dagger (y^T)^\dagger y^\dagger y$$
$$= (G^T)^\dagger (y^T)^\dagger y^T (y^T)^\dagger$$
$$= (G^T)^\dagger (y^T)^\dagger$$
$$= ((Gy)^T)^\dagger$$
$$= Gy((Gy)^T Gy)^{-1}$$
$$= ||Gy||^{-2} Gy$$
$$= sGy$$

$\square$

**Remark 1.** *When $d_2 \geq d_1$, we can construct $A$ by selecting $d_1$ orthonormal column vectors such that $A^T A = I$.*

## D MODEL ARCHITECTURE AND IMPLEMENTATION DETAILS

We evaluated the performance of FTP, BP, DTP, and PEPITA across three model families–fully connected networks (FC), convolutional neural networks (CNN), and recurrent neural networks (RNN)–on image classification and multivariate time-series forecasting tasks. While Algorithm 1 and Eqs. (1)–(4) are presented using FC networks for clarity, the underlying principles of FTP apply equally to CNNs and RNNs with only minor modifications. The core learning algorithm and update equations remain unchanged across architectures; the only adjustment required is reshaping the feedback projection to match the dimensionality of the corresponding hidden activations. For example, in CNNs, a feedback matrix $G \in \mathbb{R}^{C \times (C_1 \cdot H_1 \cdot W_1)}$ projects the output predictions or labels $y \in \mathbb{R}^{B \times C}$ into a correction matrix of shape $(B, C_1 \cdot H_1 \cdot W_1)$. This is then reshaped into a tensor of shape $(B, C_1, H_1, W_1)$, matching the spatial dimensions of the first hidden layer activations. Subsequent targets are propagated forward as in the first forward pass, and weight updates follow the same principles as in the FC case.

All models were trained using stochastic gradient descent (SGD) with momentum 0.9, batch size 64, and cross-entropy loss. Unless otherwise stated, all hidden layers used the *tanh* activation function, and all parameters (including the projection matrix $G$ in FTP) were initialized using He initialization (He et al., 2015).

### D.1 Fully Connected Networks (FC)

We evaluated FC networks on MNIST (Lecun et al., 1998), Fashion-MNIST (FMNIST) (Xiao et al., 2017), and CIFAR-10 (Krizhevsky, 2009). For MNIST and FMNIST, input images were flattened to 784 dimensions; for CIFAR datasets, to 3072 dimensions. The architecture consisted of two hidden layers with 1024 and 128 neurons, followed by a softmax output layer with 10 units for MNIST/FMIST/CIFAR-10. A dropout rate of 0.1 was applied after each hidden layer. All models were trained for 100 epochs, with the learning rate decayed by a factor of 10 at epochs 60 and 90.

### D.2 Convolutional Neural Networks (CNN)

We applied CNNs to the same four image classification datasets: MNIST (Lecun et al., 1998), CIFAR-10, and CIFAR-100 (Krizhevsky, 2009). The CNN architecture consisted of a 2D convolutional layer with 32 output channels and a 5×5 kernel, followed by a 2×2 max pooling layer. The output feature maps were flattened and passed to a softmax output layer with 10 or 100 units, depending on the dataset. All CNN models used *tanh* activations except the final layer where softmax activation function was used. The models were trained for 100 epochs, and followed the same learning rate schedule as the FC models (decayed at epochs 60 and 90).

### D.3 Recurrent Neural Networks (RNN)

We evaluated RNNs on three multivariate time-series datasets:

- *Electricity*: Electricity consumption data from 321 clients, recorded every 15 minutes from 2012 to 2014, and resampled to hourly resolution.
- *METR-LA*: Hourly road occupancy rates from 2015–2016, recorded by sensors on San Francisco Bay Area freeways.
- *Solar-Energy*: Solar power output sampled every 10 minutes during 2006 from 137 photovoltaic plants in Alabama.

Each task was structured as a sequence prediction problem, where a sliding window of the previous 24 time steps was used to predict the $25^{\text{th}}$. The RNN architecture consisted of a single recurrent layer with 512 hidden units and *tanh* activation. All RNN models were trained for 500 epochs, with the learning rate decayed by a factor of 10 at epochs 300 and 450.

## E Sensitivity to Initialization and Dimensionality of Feedback Matrix $G$

The feedback matrix $G$ in FTP plays a crucial role in projecting the local error signal (Eq. 1). In our experiments, we initialized $G$ using He Uniform (He et al., 2015), scaled by a constant factor (0.05) to regulate the strength of the feedback signal. Since $G$ directly affects the magnitude of projected error, its initialization impacts learning dynamics and gradient alignment.

To assess sensitivity to initialization schemes, we additionally trained the FC model with *He Normal* initialization (He et al., 2015), using a similar scaling factor. As shown in Table 7, FTP yields comparable performance under both schemes, suggesting that while initialization matters, the method is not overly sensitive to the exact distribution as long as the scale is appropriately controlled.

Table 7: FTP accuracy (%) under different initialization schemes for $G$. Mean ± std over 5 trials.

| Initialization | MNIST | FMNIST | CIFAR-10 |
|---|---|---|---|
| He Uniform | 97.98±0.25 | 87.24±0.21 | 52.57±0.37 |
| He Normal | 97.51±0.31 | 87.33±0.18 | 53.44±0.26 |

The dimensionality of $G$ depends on the number of output classes and the hidden layer size. As shown in Table 1, FTP achieves strong performance on both CIFAR-10 and CIFAR-100, which

differ significantly in class count (10 vs. 100), indicating generalizability across different output dimensionalities.

To further isolate the effect of class count, we reformulated MNIST into 5-class and 2-class variants (e.g., grouping digits by adjacent pairs or by even/odd labels). As shown in Table 8, FTP maintains performance comparable to BP across all class settings. These results indicate that FTP is robust to variations in $G$'s dimensionality induced by class count.

Table 8: Accuracy (%) on MNIST variants with reduced class count. Mean $\pm$ std over 5 trials.

| Method | MNIST (10) | MNIST (5) | MNIST (2) |
|---|---|---|---|
| BP | $98.27_{\pm 0.08}$ | $98.14_{\pm 0.06}$ | $98.95_{\pm 0.04}$ |
| FTP | $97.98_{\pm 0.25}$ | $97.81_{\pm 0.16}$ | $98.66_{\pm 0.12}$ |

## F  METRICS FOR EVALUATION OF RNNs

Recurrent neural networks are commonly evaluated not only by accuracy but also by statistical measures that capture both error magnitude and temporal correlation. In this work, we use Root Relative Squared Error (RRSE) and the Pearson correlation coefficient (CORR), defined as follows:

$$\text{RRSE} = \sqrt{\frac{\sum_t (y_t - \hat{y}_t)^2}{\sum_t (y_t - \bar{y})^2}}, \quad \text{CORR} = \frac{\sum_t (y_t - \bar{y})(\hat{y}_t - \bar{\hat{y}})}{\sqrt{\sum_t (y_t - \bar{y})^2} \sqrt{\sum_t (\hat{y}_t - \bar{\hat{y}})^2}} \tag{39}$$

where $y_t$ is the ground truth value, $\hat{y}_t$ is the predicted value, $\bar{y}$ and $\bar{\hat{y}}$ are the means of the ground truth and predictions, respectively.

RRSE (Root Relative Squared Error) normalizes the prediction error relative to the variance of the true series, which ensures comparability across datasets of different scales. A lower RRSE value indicates better predictive performance. CORR (Pearson correlation coefficient) measures the linear relationship between the predicted and actual sequences, capturing how well the model tracks temporal patterns. Values of CORR closer to 1 imply stronger correlation and more faithful temporal modeling. Together, these two complementary metrics provide a robust evaluation: RRSE quantifies absolute predictive accuracy, while CORR highlights alignment with the temporal dynamics of the data.

## G  SCALABILITY OF FORWARD LEARNING ALGORITHMS

Our initial experiments in Section 4.2 used a two-layer FC network with hidden dimensions of 1024–128, but under this setting PEPITA exhibited unstable training for deeper networks. To ensure a fair comparison, and following the setup in Srinivasan et al. (2023), we standardized the hidden layer size to 1024 units across all layers, allowing us to scale architectures to 2–5 layers while maintaining stability for both FTP and PEPITA. Consistent with prior observations by Srinivasan et al. (2023), PEPITA struggled to scale beyond three hidden layers: normalization improved convergence for deeper models but accuracy still declined with depth. In our experiments without normalization (Table 4), PEPITA's performance degraded as depth increased, whereas FTP remained stable and achieved accuracy close to BP without requiring normalization or additional techniques such as weight decay or weight mirroring. These results suggest that FTP scales more reliably to deeper FC networks without auxiliary mechanisms, whereas PEPITA shows reduced stability beyond shallow models. On the other hand, Forward-Forward Leearning has been limited to four hidden layers(Hinton, 2022). This points to a significant improvement over prior forward-only learning approaches.

For convolutional networks, we extended the baseline 1-Conv design by adding a second convolutional layer (3×3, 64 channels). In these settings, FTP again outperforms PEPITA, demonstrating its potential to scale effectively to CNNs.

Table 9: Test accuracy (Mean ± Std) of BP, PEPITA, and FTP on CNNs with 1 and 2 convolutional layers.

| Dataset | Algorithm | 1-Conv | 2-Conv |
|---------|-----------|--------|--------|
| MNIST | BP | $98.74_{\pm0.05}$ | $99.05_{\pm0.03}$ |
| | PEPITA | $98.41_{\pm0.24}$ | $98.35_{\pm0.05}$ |
| | **FTP** | $\mathbf{98.28}_{\pm0.35}$ | $\mathbf{98.69}_{\pm0.07}$ |
| CIFAR-10 | BP | $64.88_{\pm0.18}$ | $69.42_{\pm0.13}$ |
| | PEPITA | $56.17_{\pm0.62}$ | $57.01_{\pm0.58}$ |
| | **FTP** | $\mathbf{56.32}_{\pm0.82}$ | $\mathbf{59.57}_{\pm0.13}$ |
| CIFAR-100 | BP | $33.83_{\pm0.26}$ | $37.75_{\pm0.21}$ |
| | PEPITA | $26.77_{\pm0.87}$ | $27.61_{\pm0.20}$ |
| | **FTP** | $\mathbf{26.84}_{\pm1.13}$ | $\mathbf{29.95}_{\pm0.22}$ |

## H ROBUSTNESS OF FTP COMPARED TO FEEDBACK ALIGNMENT

To evaluate the robustness of FTP against Feedback Alignment (FA)(Lillicrap et al., 2016), we conducted experiments under the same setup as described in Section 4.6, across 10 independent trials. Unlike backpropagation, FA employs fixed random feedback matrices, eliminating the need to reprogram backward paths in hardware. Although FA is less biologically plausible than FTP, its fixed feedback makes it an important baseline for robustness under hardware constraints. We tested both 8-bit and 4-bit weight precision settings under varying levels of additive noise (0, 0.02, 0.06, 0.1). The results are reported in Table 10.

Table 10: Robustness comparison of BP, FA, and FTP under 8-bit and 4-bit precision with increasing noise levels. Accuracy reported as mean ± std over 10 trials.

| | **Noise Levels** | | | |
|---|---|---|---|---|
| **Method (8-bit)** | **0** | **0.02** | **0.06** | **0.1** |
| BP (symmetry = 100%) | $\mathbf{97.91}_{\pm0.08}$ | $\mathbf{95.60}_{\pm0.07}$ | $\mathbf{90.81}_{\pm0.17}$ | $84.56_{\pm0.81}$ |
| FA | $97.50_{\pm0.08}$ | $94.29_{\pm0.19}$ | $88.10_{\pm0.53}$ | $81.54_{\pm1.26}$ |
| FTP | $97.49_{\pm0.19}$ | $95.17_{\pm0.22}$ | $90.48_{\pm0.29}$ | $\mathbf{85.85}_{\pm0.56}$ |

| | **Noise Levels** | | | |
|---|---|---|---|---|
| **Method (4-bit)** | **0** | **0.02** | **0.06** | **0.1** |
| BP (symmetry = 100%) | $\mathbf{94.48}_{\pm0.11}$ | $90.12_{\pm0.15}$ | $81.25_{\pm1.34}$ | $11.35_{\pm0.00}$ |
| FA | $93.69_{\pm0.34}$ | $90.24_{\pm0.38}$ | $78.34_{\pm6.71}$ | $17.83_{\pm10.54}$ |
| FTP | $93.52_{\pm0.25}$ | $\mathbf{90.89}_{\pm0.27}$ | $\mathbf{82.48}_{\pm2.53}$ | $\mathbf{24.73}_{\pm16.98}$ |

As reflected in Figure 3 and the Table 10, we observe that under ideal conditions (i.e., no noise), BP and FA achieve slightly higher accuracy than FTP in low bit-precision settings. However, as noise increases, FTP demonstrates consistent and greater robustness. This suggests that FTP has inherent robustness properties that enable it to maintain stable learning dynamics under realistic hardware constraints such as low-bit precision and noise.

Importantly, since FTP uses the same forward matrices to compute both the standard activations and the layer-wise targets, any noise introduced in these matrices affects both pathways in a consistent manner. In contrast, BP requires separate programming of the forward and backward matrices in each iteration, which means programming noise can independently corrupt the two directions. This also applies for FA. This difference may have a greater impact on BP's updates compared to FTP, which may explain FTP's more stable performance in these noisy settings. In real analog hardware, additional non-idealities, such as device drift, retention loss, and other errors from the system could further exacerbate this asymmetry for BP, especially due to its reliance on reprogramming

of backward matrices. Investigating these broader effects through real hardware implementation remains an important direction for future work.

## I    MEMORY AND RUNTIME EFFICIENCY OF FTP

We analyze the memory usage and runtime cost of FTP in comparison to BP and DFA (Nøkland, 2016), particularly in the context of resource-constrained hardware.

**Memory Usage.**    The memory footprint during training arises from two primary sources: (i) activation storage during the forward pass, and (ii) storage of trainable parameters and feedback matrices.

*Activation storage:* All methods (BP, DFA, FTP) require retaining activations during the forward pass until error signals or targets are computed. Thus, peak memory usage for storing activations is comparable across all three methods.

*Parameter storage:* BP stores only the feedforward weights. FTP stores the same feedforward weights along with a single feedback matrix $G$. DFA, on the other hand, maintains a separate fixed feedback matrix for each hidden layer, resulting in the largest weight memory footprint. Notably, in deeper networks, the overhead of storing a single $G$ in FTP becomes negligible compared to the total number of trainable parameters.

**Runtime Efficiency.**    FTP performs a standard forward pass, computes the final target, and then recursively generates intermediate layer targets while updating weights layer-by-layer. Unlike BP, which updates layers in reverse order and enforces update locking, FTP permits pipelined execution. Specifically, once the first layer's weights are updated and the next target is being computed, the subsequent minibatch can begin its forward pass. This enables partial overlap between batches, potentially reducing wall-clock training time–particularly in hardware implementations that support pipelining.

DFA computes all layer gradients in parallel using fixed feedback matrices after the forward pass. The inter-minibatch delay in DFA corresponds to the time required to update all weights simultaneously. In contrast, for FTP, the delay is primarily determined by the computation of the first target and update of the first layer. Therefore, the runtime of FTP is comparable to DFA with a slight overhead for target computation, and more efficient than BP when pipelined.

## J    IMPACT OF STRICT WEIGHT SYMMETRY REQUIREMENT OF BP IN EMERGING ANALOG HARDWARE

Backpropagation (BP) relies on backward matrices to propagate errors, which must maintain symmetry with the forward weight matrices in principle mathematically, which is easily achievable in the case of digital computers. However, in analog computing hardware such as RRAM and other non-volatile memories, non-idealities such as programming errors, thermal noise, and random telegraph noise can disrupt this symmetry, leading to significant degradation in performance. Our results, as shown in Figure 4, demonstrate the vulnerability of BP under these conditions, particularly in devices with low-bit precision. Here we considered the cases when some weight parameters in a matrix got corrupted due to device non-idealities, and the asymmetry here refers to the amount of weights getting corrupted by 10% margin from where the values should be. For instance, in 4-bit systems (Zidan et al., 2018; Yao et al., 2020), test accuracy drops below 80% when only 20% of the matrices are affected by such errors. The vulnerability is even more pronounced in 3-bit systems, where a mere 5% mismatch between forward and backward matrices causes test accuracy to plummet below 60%. As asymmetry increases, test accuracy steadily declines, eventually reaching approximately 15%, which is equivalent to random guess. The standard deviation of test accuracy also increases, which indicates greater instability in performance. Each case was tested multiple times, and the shadowed region in Figure 4 represents the standard deviation of test accuracy across trials. In contrast, FTP utilizes fixed, random backward matrices for target estimation, eliminating the dependency on symmetric forward-backward matrix relationships. This design inherently makes FTP

resilient to hardware-induced noise and non-idealities, ensuring robust performance even in low-bit precision environments.

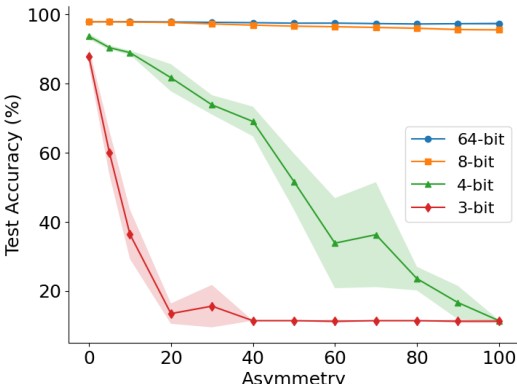

Figure 4: Performance of BP when asymmetry is introduced in backward matrices due to read noise.

## K  COMPUTE RESOURCES

All experiments were conducted on a local workstation configured with the following specifications: an AMD Ryzen Threadripper PRO 5955WX processor (16 cores, 4.00–4.50GHz), 256 GB of DDR4-3200 RAM, and dual NVIDIA GeForce RTX 4090 GPUs.

Experiments involving CNNs and RNNs were executed on the GPUs to leverage accelerated training, while fully connected network experiments and all evaluations in Section 4.4, Section 4.5 and Section 4.6 were performed on CPU only.

## L  USE OF LARGE LANGUAGE MODELS (LLMS)

Large language models (LLMs) were used in this work solely as a general-purpose assistive tool for grammar checking and text editing. All research ideas, experimental designs, analyses, and scientific claims were conceived, implemented, and validated solely by the authors.

