# OpenReview forum: "Forward Target Propagation: A Forward-Only Approach to Global Error Credit Assignment via Local Losses"
_ICLR.cc/2026/Conference — ICLR 2026 Conference Withdrawn Submission_

### Official Review · Reviewer_VvXt · 2025-10-20

**Soundness:** 3
**Presentation:** 3
**Contribution:** 3
**Rating:** 6
**Confidence:** 4

**Summary:**

The paper proposes Forward Target Propagation (FTP), a forward-only training rule that replaces the backward pass with a second forward pass driven by layer-wise “targets.” Concretely, FTP forms a first-layer target by contrasting projected labels and current outputs through a fixed random matrix G and adds this correction to the first hidden activation; deeper targets are then obtained by reusing the standard feedforward weights, and each layer minimizes a local loss to its target. The authors claim FTP avoids symmetric weight transport, alleviates update locking via pipelining, aligns its updates with BP, and is more robust/efficient than several biologically inspired alternatives. Experiments on small FC/CNN models (MNIST, FMNIST, CIFAR-10/100) and an RNN for time-series forecasting show FTP performs close to BP and better than DTP/PEPITA on some axes, with MAC counts near BP and improved robustness under quantization/noisy hardware.

**Strengths:**

1. Clear, simple forward-only rule with an easily implementable target construction and a single fixed feedback matrix.

2. The method extends to RNNs with minimal changes, and the study includes multivariate time-series forecasting, which shows the potential of adapting to various domains.

**Weaknesses:**

1. All experiments are conducted on small datasets, not large-scale ones. The reviewer is curious about the performance on large-scale datasets such as ImageNet.

2. The experiment lacks some recent baselines. The reviewer is wondering how FTP performs compared to these models.[1-3]

3. No direct runtime measurements or memory traces to substantiate efficiency beyond MAC estimates. The reviewer wants to see the training time, RAM usage, etc., compared with other baselines.

[1] Kappel, David, Khaleelulla Khan Nazeer, Cabrel Teguemne Fokam, Christian Mayr, and Anand Subramoney. "A variational framework for local learning with probabilistic latent representations." In 5th Workshop on practical ML for limited/low resource settings.

[2] Zhang, Aozhong, Zi Yang, Naigang Wang, Yingyong Qi, Jack Xin, Xin Li, and Penghang Yin. "Comq: A backpropagation-free algorithm for post-training quantization." IEEE Access (2025).

[3] Cheng, Anzhe, Heng Ping, Zhenkun Wang, Xiongye Xiao, Chenzhong Yin, Shahin Nazarian, Mingxi Cheng, and Paul Bogdan. "Unlocking deep learning: A bp-free approach for parallel block-wise training of neural networks." In ICASSP 2024-2024 IEEE International Conference on Acoustics, Speech and Signal Processing (ICASSP), pp. 4235-4239. IEEE, 2024.

**Questions:**

Please see the weakness below.

---

### Official Review · Reviewer_CkJz · 2025-10-30

**Soundness:** 4
**Presentation:** 4
**Contribution:** 3
**Rating:** 6
**Confidence:** 4

**Summary:**

This paper introduces FTP as an alternative to backpropagation. Results show competitive performance, also outperforming other related algorithms for effective learning.

**Strengths:**

This paper provides a promising alternative to backpropagation for neural network training. Results show competitive performance compared to other related algorithms and learning stability for mid-sized networks. The appendices provide extensive additional detail about the behaviour of the algorithm as well as a theoretical derivation of the approach.

**Weaknesses:**

The authors provide a scaling experiment. However, scaling is up to max 5 layers and only 2 convlayers. This leaves me wondering how the algorithm behaves for much larger networks. It would be relatively easy to test this. E.g. how does it perform wrt architectures like very deep Resnets trained on imagenet? Does it reach SOTA accuracy? How about LLMs and other architectures? The reason I am asking is because many alternatives to BP tend to break down at much larger scales. It is important to analyse this in more detail, e.g. by training resnets of different size (say up to 50 layers) and show differences with BP. This analysis would either show when FTP breaks down or demonstrate effective learning on modern architectures. This scaling analysis is an important addition. Instead of demonstrating SOTA performance on challenging tasks with deep architectures (preferred) and alternative may be to analyse BP vs FTP when increasing the number of layers on the current tasks, to showcase that even for very deep architectures, performance is maintained. If this scaling analysis is provided, I am willing to increase my rating.

There is related work which shows that target propagation is equivalent to backpropagation under orthogonal weight matrices. I am wondering if the authors can claim that FTP is equivalent in the absence of such constraints. See:
https://proceedings.neurips.cc/paper/2020/file/7ba0691b7777b6581397456412a41390-Paper.pdf

**Questions:**

Line 194 and beyond: make h_i boldface

Wrt the claim about biological plausibility one nuance could be added that real neurons learn based on spike timing.

Where do the MAC differences come from?

Ensure that the use of boldface is consistent across tables. Don’t use it to denote your algorithm but to indicate best performance. If you do, then do this across all tables.

Since gamma is so crucial; how was gamma used to produce the main results? Why not optimize there as well?

Fig 2: the angle progression seems somewhat parabolic in shape. Why is this the case? What happens when the network is trained for a very long time? Do we see double descent phenomena? Does the angle difference go to zero? The latter would indicate that a local optimum is found.

You mention that FTP is a partial resolution to update locking. Why ‘partial’? Explain what is still missing.

Line 1022: Fix Leearning; add whitespace before citation.

The approach as such is a modification of existing approaches like PEPITA. I am not 100% sure if this the modification is not already identified in the literature. The authors may want to comment on how FPT is a nontrivial extension of existing work.

---

> ### Comment · Reviewer_CkJz · 2025-11-28
>
> Given the concerns on scaling and overlap with related work by the other reviewers and the absence of an author response I am changing my rating to 4 (marginally below acceptance threshold).

---

### Official Review · Reviewer_WrCd · 2025-10-31

**Soundness:** 2
**Presentation:** 3
**Contribution:** 2
**Rating:** 4
**Confidence:** 4

**Summary:**

The article proposes an alternative to backpropagation (BP) intended both as a biologically plausible learning rule and as an efficient machine-learning method. The approach is forward-only, with no explicit backward pass during training.

The algorithm has two phases:

1. Inference: The input is propagated through the forward pathway to produce the output.

2. Training: The output and labels are projected to the first hidden layer to form a target signal for that layer, after which a second forward pass is applied to this signal.

Conceptually, the method blends ideas from forward-only algorithms like PEPITA—where the network’s second pass takes the output as input—and from difference target propagation in terms of target generation method (for the first hidden layer).

The article includes numerical experiments comparing the proposed method with other biologically plausible approaches.

**Strengths:**

- The idea is new (to my knowledge), even though it combines elements from existing target-propagation and forward-only methods.

- The algorithm appears satisfactory with respect to several goals: biological plausibility, hardware-efficient implementation, and robustness.

**Weaknesses:**

- The core idea lacks analytical justification; the paper offers no convincing theoretical reasoning and relies primarily on empirical evidence.

- The experiments are not compelling: network depths and sizes are small (only one or two hidden layers are used).

**Questions:**

- Can the method be derived more principledly—for example, as an approximation to BP?

- What is the impact of the choice of $G$?

- Is forward target propagation (FTP) scalable to deep networks and high-dimensional inputs?

---

### Official Review · Reviewer_K3T6 · 2025-10-31

**Soundness:** 1
**Presentation:** 2
**Contribution:** 2
**Rating:** 2
**Confidence:** 4

**Summary:**

This paper proposes Forward Target Propagation (FTP) as an alternative to backpropagation. It computes a first-layer target by projecting both the label and the current output through a fixed random matrix and nonlinearity, which is then propagated forward through the usual feedforward weights. The weights are updated with local losses encouraging alignment between activations and targets. Experiments on small FC/CNN/RNN models for image and time-series datasets verify the effectiveness of the method, with additional simulations of low-precision/noisy “analog” hardware.

**Strengths:**

1. The core mechanism is easy to implement and reason about. It avoids the symmetric weight transport, alleviates update locking, and aligns with BP directions.
2. The paper include hardware-motivated discussions and show the robustness under various settings.

**Weaknesses:**

1. Core claims are severely overstated or imprecise.

1.1) “Forward-only” is misleading. Although gradients are not propagated backward, FTP does use a top down feedback pathway (the fixed matrix projecting from output space to the first hidden layer, Fig. 1d). It is more similar to DFA-series works compared to “forward-only” methods.

1.2) “Local credit assignment” is imprecise. The method does rely on global signals (top-down feedback from output space) and requires temporal non-locality to wait for propagation to the output space and then feedback.

1.3) “Solves update locking partially” is imprecise. The training still waits for a full forward propagation and top-down feedback, as well as the second round of forward propagation. It cannot solve update locking.

2. Experiments and baselines are too toy to support scalability claims. The models considered in the paper are small and have very low accuracies on standard datasets such as CIFAR-10/100, missing standard modern baselines or large-scale datasets.

3. Insufficient comparisons. The method is only compared to BP and PEPITA, missing other more bio-plausible algorithms such as DFA etc.

**Questions:**

See Weaknesses.

---

### Note · Authors · 2025-12-04

I have read and agree with the venue's withdrawal policy on behalf of myself and my co-authors.